# Measuring Digital Citizenship: A Comparative Analysis

Juan Sebastián Fernández-Prados [1], Antonia Lozano-Díaz [2,*] and Alexandra Ainz-Galende [1]

[1] Department of Sociology, University of Almería, CEMyRI, 04120 Almería, Spain; jsprados@ual.es (J.S.F.-P.); aag486@ual.es (A.A.-G.)

[2] Department of Education, University of Almería, 04120 Almería, Spain

\* Correspondence: ald805@ual.es; Tel.: +34-950-015-968

**Abstract:** This paper aims at showing a state of the art about digital citizenship from the methodological point of view when it comes to measuring this construct. The review of the scientific literature offers at least ten definitions and nine different scales of measurement. The comparative and diachronic analysis of the content of the definitions shows us two conceptions of digital citizenship, some more focused on digital competences and others on critical and activist aspects. This paper replicates and compares three scales of measurement of digital citizenship selected for their relevance and administered in a sample of 366 university students, to analyze their psychometric properties and the existing coincidences and divergences between the three. The most outstanding conclusion is that not all of them seem to measure the same construct, due to its diversity of dimensions. An online activism dimension needs to be incorporated if digital citizenship is to be measured. There is an urgent need to agree internationally on a definition of digital citizenship with its corresponding dimensions to elaborate a reliable and valid measuring instrument.

**Keywords:** citizen participation; comparative analysis; measurement technique; digital citizenship; online activism





## 1. Introduction

Academic publications, and particularly international educational practices and programs, have shaped different ideas and approaches to digital citizenship since its start [1]. Authors have gone even further by adding other epithets such as democratic digital citizenship [2], global digital citizenship [3], radical digital citizenship [4], etc. This process has been reflected in an evolution that, in the words of Heath [5], has shifted from conservative and more technologically constrained positions to more critical and active ones (see Table 1).

Thus, the concept of digital citizenship has evolved from one of the first definitions of digital citizenship written by Ribble and Bailey [6], which focused on technological aspects and digital competencies, to the one proposed by Emejulu and McGregor [4], which highlights the commitment to social justice and to emancipatory and alternative technology. In the same vein, Pangrazio and Sefton-Green [7] point out that early conceptions of digital citizenship were concerned with the individual's right to access and participate online to bridge the digital divide. Currently, the relationship of citizenship with the digital world has become much more complex around collective identities and in the context of social networks with ample possibilities.

This recent trend and conceptual process of the category of digital citizenship shows the diversity of approaches and perspectives, whether they are educational, socio-political, etc. For example, Law, Chow and Fu [8] consider up to three curricular or pedagogical perspectives of digital citizenship, such as digital competence, as an integral part of the culture of information and a final perspective related to the preparation for civic participation and politics. Likewise, Ryland [9] places the different approaches and definitions of digital citizenship into four groups: digital literacy, digital access, digital character, and

civic engagement. It is not surprising that the most recent publications call for the urgent redesign and redefinition of digital citizenship to include the latest contributions of the most advanced, critical, and global concepts, as well as to consider the broader contexts and the latest educational practices [10,11].

**Table 1.** Scales for measuring digital citizenship and psychometric properties.

| Source | Definition |
|---|---|
| Ribble and Bailey, 2007 [6] (p. 10) | "norms of appropriate, responsible behavior with regard to technology use. Digital citizenship Is a concept which helps teachers, technology leaders, and parents to understand how use technology appropriately" |
| International Society for Technology in Education, 2008 [12] (p. 1) | "advocate and practice safe, legal, and responsible use of information and technology; exhibit a positive attitude toward using technology that supports collaboration, learning, and productivity; demonstrate personal responsibility for lifelong learning; exhibit leadership for digital citizenship." |
| Mossberger, Tolbert, and McNeal 2008 [13] (p. 1–2) | "those who use the internet regularly and effectively- that is, on a daily basis [ . . . ] digital citizens are those who use technology frequently, who use technology for political information to fulfill their civic duty, and who use technology at work for economic gain" |
| Robles, 2009 [14] (p. 55) | "that individual, citizen or not of another community or State, who exercises all or part of his political or social rights through the Internet, independently or through his membership in a virtual community" |
| Ohler, 2010 [15] (p. 187) | "I can make the topic much more accessible if I refer to digital citizenship as "character education for the Digital Age." |
| Richards, 2010 [16] (p. 518) | "practices conscientious use of technology, demonstrates responsible use of information, and maintains a good attitude for learning with technology" |
| Choi, 2016 [2] (p. 565) | "4 major categories that construct digital citizenship: Ethics, Media and Information Literacy, Participation/Engagement, and Critical Resistance." |
| eTwinning, 2016 [17] (p. 11) | "Three main pillars come to mind when trying to define digital citizenship: belonging, engagement, and protection. Digital citizens belong to the digital society. They use technology to actively engage in and with society. Digital citizenship empowers people to reap the benefits of digital technology in a safe and effective way." |
| Council of Europe, 2017 [18] (p. 10) | "Digital Citizenship may be said to refer to the competent and positive engagement with digital technologies and data (creating, publishing, working, sharing, socializing, investigating, playing, communicating and learning); participating actively and responsibly (values, skills, attitudes, knowledge and critical understanding) in communities (local, national, global) at all levels (political, economic, social, cultural and intercultural); being involved in a double process of lifelong learning (in formal, informal, non-formal settings) and continuously defending human dignity and all attendant human rights" |
| Emejulu and McGregor, 2019 [4] (p. 140) | "as a process by which individuals and groups committed to social justice deliberate and take action to build alternative and emancipatory technologies and technological practices" |

Source: Prepared by authors.

A review of literature using Google Scholar and Scopus database provides at least ten research projects that have had, among other results, the construction of a scale or instrument to measure digital citizenship ("Digital citizenship scale"; Table 2). The publications come from a diverse range of countries, although there are more from Asia and the Middle East (China, Malaysia, South Korea, Jordan, Saudi Arabia, Oman, Jordan, and Turkey) than from the West (USA, Canada, and Spain). A search in the databases also provides the number of quotes of each category that has any of the ten scales found, with 151 mentions in Google Scholar and 58 in Scopus by Jones and Mitchell [19] followed with 90 and 27 quotes, respectively, by Choi, Glassman, and Cristol [20].

**Table 2.** Scales for measuring digital citizenship and psychometric properties.

| Authors, Year—Country [Source] (Citations Google Scholar/Scopus) | Sample (Items—Options) Cronbach's Alpha | Dimensions |
|---|---|---|
| Isman and Gungoren, 2014—Saudi Arabia [21] (81/16) * | 229 college students (34 items—5 opt.) α = 0.85 | 1. Digital Literacy<br>2. Digital Law<br>3. Digital Rights and Responsibility<br>4. Digital Communication<br>5. Security Digital<br>6. Digital Commerce<br>7. Digital Access<br>8. Digital Etiquette<br>9. Digital Health and Wellness |
| Al-Zahrani, 2015—Saudi Arabia [22] (50/14) * | 174 college students (46 items—5 opt.) α = 0.92 | 1. Respect Yourself/Others<br>2. Educate Yourself/Others<br>3. Protect Yourself/Others |
| Nordin et al., 2016—Malaysia [23] (23/4) * | 391 college students (17 items—5 opt.) α = 0.78 a 0.86 | 1. Etiquette<br>2. Responsibility<br>3. Wellbeing/Health<br>4. Commerce<br>5. Security |
| Jones and Mitchell, 2016—USA [19] (151/48) * | 979 high school students (11 items—5 opt.) α = 0.70 and α = 0.92 | 1. Online Respect<br><br>2. Online Civic Engagement |
| Choi, Glassman, and Cristol, 2017—USA [20] (90/27) * | 508 college students (26 items—7 opt.) α = 0.88 | 1. Internet Political Activism<br>2. Technical Skills<br>3. Local/Global Awareness<br>4. Critical Perspective<br>5. Networking Agency |
| Torrent-Sellens and Martínez-Cerdá, 2017—Spain [24] (8/2) * | 544 college students (8 items—5 opt.) α = 0.94 | 1. "Ciudadanía activa" (5 items, α = 0.93)<br>2. "Uso diversificado de los medios" (3 items, α = 0.95) |
| Hui and Campbell, 2018—Canada [25] (12/2) * | 26 college students (40 items—7 opt.) α = 0.80 | Idem . . . [20] |
| Mata-Domingo and Guerrero, 2018—Oman [26] (3) | 200 high school students (9 items—4 opt.) | Idem . . . [21] |
| Jwaifell, 2018—Jordan [27] (6) | 263 college students (9 items—2 opt.) | Idem . . . [20] |
| Peart, Gutiérrez-Esteban, and Cubo-Delgado, 2020—Spain [28] (0/0) * | 205 secondary schools, universities and NGOs ** (59 items—5 opt.) α = 0.91 and α = 0.90 | 1. Digital Skills<br>2. Socio-civic Skills |

Source: Prepared by authors, based on Lozano-Díaz and Fernández-Prados [15] (p. 85). * Indexing in Scopus. ** Non-Governmental Organization.

The next column of Table 2 provides information on the recipients to whom the instrument was administered in its first application, mostly to university students or recent graduates [21–23], to adolescent or secondary school students [19,26,28], and to teachers [25,27]. Some technical and psychometric properties of the scales are also shown, such as the samples, small and random in all cases; characteristics of the instruments, between 8 and 59 items and between 2 and 7 response options; and the reliability of the scales using Cronbach's alpha, always above 0.70.

Finally, the third column contains the list of dimensions of each scale, between two and nine, resulting chiefly from the application of a theoretical model or concept such as the three mentioned above [21,25,27], which literally apply the approach of Ribble and Bailey's nine elements [21], without performing a critical analysis on the theoretical corpus to date or considering the creation and selection procedure of items by judges and experts [2,20]. Only some scales apply an exploratory and confirmatory factor analysis to obtain their dimensions [23,24].

Only five scales have been replicated in the last few years by diverse authors (see Table 3). Thus, the most replicated scale, six times, has so far been that of Choi Glassman and Cristol, while Nordin et al. have replicated it only once. In addition to this diversity in the repercussion and impact on the scientific community, the authors of the replications have made small variations in the number of questions or response options. These studies have also expanded the number of countries (China, Turkey, South Korea, etc.) and target groups (teachers, college students, high school, etc.). In any case, they have ratified and validated the instruments in other cultural contexts and have expanded the scientific and experimental corpus.

**Table 3.** Replications or adaptations of main digital citizenship scales.

| Authors, Year [Source] | Authors, Year—Country [Source] | Sample (Items—Options) | |
|---|---|---|---|
| | | Cronbach's Alpha (Dimensions) | |
| Isman and Gungoren, 2014 [21] | Aladag and Ciftci, 2017—Turkey [29] | 346 teachers (34 items—5 opt.) (9 dimensions) | |
| | Elcicek, Erdemci, and Karal, 2018—Turkey [30] | 143 college students (34 items—5 opt.) $\alpha = 0.79$ (9 dimensions) | |
| | Elmali, Tekin, and Polat, 2020—Turkey [31] | 80 teacher candidates (34 items—5 opt.) $\alpha = 0.85$ (9 dimensions) | |
| Al-Zahrani, 2015 [22] | Alqahtani, Alqahtani, and Alqurashi, 2017—USA [32] | 51 college students (46 items—5 opt.) $\alpha = 0.91$ (3 dimensions) | |
| | Alqahtani, 2017—Saudi Arabia [33] | 361 teachers (46 items—5 opt.) $\alpha = 0.89$ (3 dimensions) | |
| | Ke and Xu, 2017—China [34] | 115 college students (46 items—5 opt.) (3 dimensions) | |
| | Xu, Yang, MacLeod, and Zhu, 2019—China [35,36] | 712 college students (46 items—5 opt.) $\alpha = 0.89$ (3 dimensions) | |
| Nordin et al., 2016 [23] | Jwaifell, Aljazi, and Gasaymeh, 2019—Jordan [37] | 189 high school students (16 items—5 opt.) | |
| Choi, Glassman, and Cristol, 2017 [20] | Kim and Choi, 2018—South Korea [38] | 200 teachers (18 items—7 opt.) $\alpha = 0.75$ (5 dimensions) | |
| | Kara, 2018—Turkey [39] | 435 college students (26 items—7 opt.) $\alpha = 0.89$ (5 dimensions) | |
| | Choi, Cristol, and Gimbert, 2018—USA [40] | 348 teachers (26 items—7 opt.) $\alpha = 0.79$ a 0.89 (5 dimensions) | |
| | Erdem and Koçyiğit, 2019—Turkey [41] | 272 college students (26 items—7 opt.) $\alpha = 0.87$ (5 dimensions) | |
| | Yoon, 2019—South Korea [42] | 283 college students (23 items—5 opt.) $\alpha = 0.88$ (5 dimensions) | |
| | Lozano-Díaz and Fernández-Prados, 2020—Spain [43,44] | 302 college students (26 items—7 opt.) $\alpha = 0.89$ (5 dimensions) | |

Source: Prepared by authors.

Given this amalgam of definitions and measurement instruments of digital citizenship, this main aim of this paper is to replicate and compare the most important scales and to discuss the weaknesses and strengths of each. The conclusions obtained can be used to make headway in this recent field of research [45] with an ever-growing interest in education and social sciences around the world [46].

## 2. Materials and Methods

### 2.1. Materials and Scales

First, three of the ten scales found in the review were selected. The selection criteria fundamentally responded to the impact on academic literature and citations in Scopus. As a result, the article by Jones and Mitchell scale (JM) [19] was the most quoted, with 48 publications on Scopus; the Choi, Glassman, and Cristol (CGC) [20], which was the second most quoted, had 27 references; and finally, Al-Zahrani's scale [22] was selected despite having a somewhat smaller number of quotations than Isman's [21] because it is more reliable both in its scale and in the numerous replications.

An expert translated digital citizenship scales from English to Spanish (see Appendix A). Prior to their application with the sample participants, a test was conducted with ten students to detect errors of comprehension or of another type, thereby making it possible to

carry out the necessary adjustments. The questionnaire with the scales was administered over the internet using the open source LimeSurvey, an application for online surveys (for more information, see the LimeSurvey website: the online survey tool open-source surveys. URL: https://www.limesurvey.org, accessed on 24 January 2021). It was sent to the participants of the sample by email, while a brief explanation was given in class beforehand on how to respond and self-administer said scale.

### 2.2. Sample

The sample was selected among students from one university, like most of the studies referenced in Table 2, during the 2019–2020 academic year. It involved a sample of 366 students out of a total of 12,000 at the university, belonging to various schools such as the School of Education (Degree in Early Childhood Education, Primary Education and Social Education), the School of Law and the School of Social Work [47]. Thus, for this population and sample as well as for a confidence level of 95%, the sampling error was ±5%. The sample profile by sex was 78.7% female and 21.3% male, and by age, the majority was between 18 and 22 years old (79.5%).

### 2.3. Analysis

The resulting data analyses set out to compare the psychometric properties of each scale, which is why we performed a reliability analysis with Cronbach's alpha statistic, and we performed an exploratory factor analysis by extracting the principal components to obtain the dimensions and the total explained variance. It also set out to establish the possible relationship among the three scales and their dimensions through correlation with Spearman's rank correlation coefficient. Likewise, after categorizing the scales into high, medium, and low digital citizenship by dividing the distribution into terciles, they were compared in order to estimate the coincidences and the chi-squared statistic. In summary:

- Analysis of the psychometric properties of each scale
- Correlational analysis between the scales and their dimensions
- Comparative analysis of the digital citizenship level classifications of the scales and their coincidences

### 2.4. Limitations

The first limitation of this study is the sample drawn from a university context. All the studies reviewed, like this one, selected their sample randomly or incidentally from an educational group. This decision configures samples that are excessively homogeneous by age and educational level.

The second limitation is the difficulty of administering questionnaires developed by other researchers in different countries. Both the translation and the cultural contexts of different countries can lead to errors and make comparisons difficult.

In addition to the methodological limitations mentioned above and others, the diversity of definitions, the different theoretical approaches, and the evolution of the term "digital citizenship" also make it difficult to find coincidences among measurement instruments.

## 3. Results

### 3.1. Analysis of the Psychometric Properties of Each Scale

The result of the main psychometric properties of the three scales achieved more than sufficient scores about the reliability of the measurement instruments (see Table 4). Cronbach's alpha coefficient exceeded 0.80 in all three cases, although the analysis of the correlation of each item with the total of its scale would question its presence, as it was below 0.30 for eight items on the Al-Zahrani (AZ) scale and four items on the Choi, Glassman, and Cristol (CGC) scale.

**Table 4.** Psychometric properties of the three digital citizenship scales (DCS).

| DCS | Reliability Analysis [1] | KMO and Bartlett's Test [2] | Factor Analysis and Principal Component Analysis [3] |
|---|---|---|---|
| AZ scale | 0.89 (8) | 0.80 ($p < 0.001$) | 12 components and 62.73% |
| JM scale | 0.83 (0) | 0.84 ($p < 0.001$) | 2 components and 54.18% |
| CGC scale | 0.90 (4) | 0.84 ($p < 0.001$) | 6 components and 69.85% |

[1] Cronbach's alpha (numbers of items with corrected item-total correlation < 0.30). [2] Kaiser–Meyer–Olkin (KMO) measure of sampling adequacy (Bartlett's test of sphericity and *p*-values). [3] Numbers of components and total variance explained (fixed number of factors by original scale and total variance explained). AZ, Al-Zahrani scale; JM, Jones and Mitchell scale; CGC, Choi, Glassman, and Cristol scale. Source: Prepared by authors.

Meanwhile, the Kaiser–Meyer–Olkin (KMO) measure was high and significant enough with Bartlett's test of sphericity to propose, in the three scales, an exploratory factor analysis to extract the main components. Therefore, from the principal components we extracted all the factors whose eigenvalues exceeded one to compare them with the fixed number that the corresponding authors proposed in their respective papers.

The AZ scale was far from the number of components extracted; in fact, if the factor analysis is attached to the three proposed components, it loses a great deal of the explanatory power of the variance. Meanwhile, in the JM scale the number of components matched a total explained variance of more than 50%. Finally, in the CGC scale only one more component appeared, yet it was the instrument that explained the most variance (66%). In summary, the AZ digital citizenship scale, despite its high reliability, raised severe doubts about the constitution of its components based only on the exploratory factor analysis, while the JM and CGC scales, which were both very reliable, showed a better match of their components and obtained a high explained variance, especially the CGC.

*3.2. Correlational Analysis between the Scales and Their Dimensions*

Table 5 shows, in the first column, the relationship of the various components of the scales as defined by the authors. Therefore, the three dimensions of the AZ scale exceeded the minimum reliability of 0.70, while one of the two components of the JM scale did not reach the indicated threshold, nor did one of the five of the CGC scale. At this point, we should recall that the AZ instrument was the one that included the greatest number of items, which helps obtain better coefficients.

The Spearman rank correlation coefficient and the significance level of all dimensions with the three digital citizenship scales are also recorded in the following columns. As expected, the dimensions of each scale correlated highly with the total of its questionnaire (with a significance level of $p < 0.001$). One dimension stood out in each one of them: in the AZ scale, the component "Respecting oneself and others"; in the JM scale, "Online respect"; and in the CGC scale, two dimensions, "Internet political activism" and "Critical approach", which, to a certain extent, defined the emphasis or focus of the instrument.

The two scales that most correlated with each other in their dimensions and in the total were the AZ scale with the JM scale, doing so at a level of 0.30, which was significant ($p < 0.001$), as can be seen in the last three rows of Table 4. Meanwhile, the AZ and CGC scales did not correlate as a whole or by dimensions, except one of AZ with the CGC scale. Again, it was the JM scale that correlated with an index of 0.23 ($p < 0.001$) with the CGC scale.

**Table 5.** Correlations among the different dimensions of the CD scales.

| DCS, Dimensions. (Cronbach's Alpha) | AZ Scale | JM Scale | CGC Scale |
|---|---|---|---|
| AZ scale | 1.000 | 0.295 *** | 0.093 |
| Dimension 1. Respect Yourself/Others (0.87) | 0.853 *** | 0.274 *** | 0.023 |
| Dimension 2. Educate Yourself/Others (0.72) | 0.647 *** | 0.263 *** | 0.348 *** |
| Dimension 3. Protect Yourself/Others (0.86) | 0.768 *** | 0.206** | 0.042 |
| JM scale | 0.295 *** | 1.000 | 0.233 *** |
| Dimension 1. Online Respect (0.85) | 0.249 ** | 0.905 *** | 0.100 |
| Dimension 2. Online Civic Engagement (0.67) | 0.248 ** | 0.804 *** | 0.347 *** |
| CGC scale | 0.093 | 0.233 *** | 1.000 |
| Dimension 1. Internet Political Activism (0.91) | 0.158 * | 0.154 * | 0.820 *** |
| Dimension 2. Technical Skills (0.92) | −0.025 | 0.167 ** | 0.207 *** |
| Dimension 3. Local/Global Awareness (0.79) | 0.010 | 0.153 * | 0.488 *** |
| Dimension 4. (0.84): Critical Perspective | 0.035 | 0.203 ** | 0.863 *** |
| Dimension 5. Networking Agency (0.64): | 0.043 | 0.141 * | 0.704 *** |

\* $p < 0.05$. ** $p < 0.01$. *** $p < 0.001$. Source: Prepared by authors.

### 3.3. Comparative Analysis of the Digital Citizenship Scales

Finally, the three scales were compared according to coincidences in classifying distinct levels of digital citizenship. Each scale was categorized into terciles of their respective distributions—in other words, three levels of high, medium, or low digital citizenship. Therefore, Figure 1 shows that the three scales coincided in classifying 22% of respondents in the same category or level. In addition, the Venn diagram shows the percentage of respondents classified in the same category by pairs of scales; for example, those that shared the most, the JM and AZ scales, were the ones that coincided the most (22.2% + 29.4% = 51.6%). The scale that shared the least with the other two was CGC (with only 61.1%), and the scale that shared the most with the other two was JM (with 73.8%).

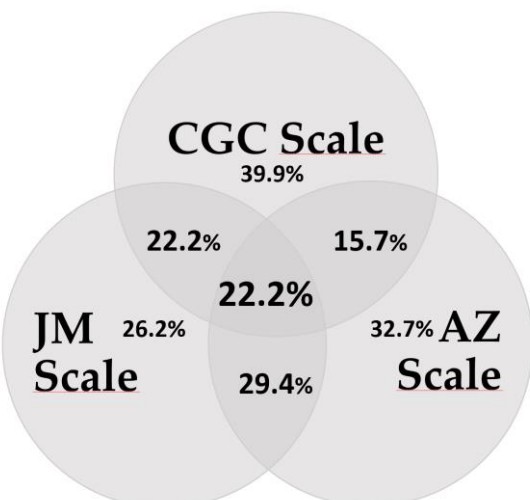

**Figure 1.** Venn diagram with coincidence percentages among scales.

Likewise, the three scales organized in the three categories or levels were crossed with each other to calculate the chi-square statistic (see Table 6). It can be seen how the Jones and Mitchell (JM) with the Al-Zahrani scale (AZ) obtained a statistically highly significant chi-square ($p < 0.001$), while the AZ and CGC scales were far from being statistically significant.

**Table 6.** Percentage of the sample in the same terciles [chi-square test].

| DCS in Terciles | JM Scale | CGC Scale |
|---|---|---|
| AZ scale | ($\chi^2$ (4, $n$ = 153) = 28.12 $p$ < 0.001) *** | ($\chi^2$ (4, $n$ = 153) = 4.45 $p$ = 0.346) |
| JM scale | - | ($\chi^2$ (4, $n$ = 153) = 15.06 $p$ < 0.01) ** |

** $p$ < 0.01. *** $p$ < 0.001. Source: Prepared by authors.

## 4. Discussion

The introduction already warned of the diversity of definitions and the variety of instruments for measuring digital citizenship that required a methodological approach with a comparative perspective, replicating some of the most prominent scales. The most prominent results point to methodological deficiencies in the Al-Zahrani scale when specifying the components of its instrument and not providing a factor analysis, neither exploratory nor confirmatory, to support them statistically. Likewise, correlation and statistical analyses such as the chi-square statistic conclusively showed the scarce relationship between the scale with the most items and impact, that of Al-Zahrani, and the most replicated scale, that of Choi, Glassman and Cristol. In summary, the review, replication, and comparison lead us to two underlying issues or problems related to these findings—one of a conceptual and theoretical nature, and the other of a methodological nature.

The conceptual debate was not only triggered by a wide list of definitions on digital citizenship but by the various theoretical approaches described by different authors [7–9], which somehow recall the disquisitions of the theory policy around the two existing concepts of citizenship, liberal citizenship, and republican citizenship [14], and which required a study in itself. In any case, at least two questions should be raised for the re-elaboration or reconceptualization of digital citizenship and its application in instruments for measuring it.

First, it is necessary to eliminate the component related to digital literacy or skills, which, in addition to having its own scientific literature and differentiated foundation [38], is less and less important in a highly literate generational context and which, as seen in the Choi scale, is the component with the lowest correlation with its own scale. Secondly, it is necessary to incorporate the activist and critical component as part of the essence of digital citizenship, following not just in the wake of the latest definitions but also educational practices across the world in education for global citizenship [10,48,49]. Certainly, these components and their emphasis on involvement and participation shift the Choi, Glassman, and Cristol scale further away from the Al-Zahrani scale, which accentuates more components of protection and respect or, in other words, a more "defensive" and "negative" version of digital citizenship, inspired by the definition of Ribble and Bailey [6].

The methodological aspects and their restraints appear on a recurring basis in the creation of new instruments to study or evaluate emerging realities such as digital citizenship. The methodological problems focus on aspects such as (i) the elaboration of instruments based on definitions of specific authors without a thorough critical review; (ii) the incorporation of items with no other criteria than the own intuition of the researcher or researchers; (iii) the fact of ignoring the validation of content and of the construct of the questionnaires; (iv) the use of random samples of little variability; and (v) and the failure to give an option to access the data matrices, etc.

## 5. Conclusions

As a general conclusion and future research proposal, we should highlight the fact that more replications, comparatives, and studies are needed in the use of digital citizenship assessment tools to reach a consensus. In them, it is necessary to (i) clarify the citizenship theory model under consideration; (ii) explain which correlates can be found with instruments that claim to assess the same construct; (iii) expand the study sample as far as possible in the general population; and (iv) observe the necessary methodological rigor. At the same time, theoretical debate on the construct of digital citizenship is also urgently



needed to reach a consensus around it. In a context of international and transparent studies with widely tested questionnaires, it is necessary to move toward the design of a rigorous, reliable, and validated digital citizenship measurement instrument, one which is also administered in equiprobable and representative samples. In this sense, the first steps taken by international organizations such as the European Union, UNICEF, and UNESCO are encouraging [50–52].

This theoretical–conceptual and methodological–instrumental consensus would be particularly useful to assess not only different dimensions of students' digital competencies [53] but also other aspects and social contexts. Thus, whether for addressing challenges in small rural communities [54] or for major sustainable development goals [55], assessing the participation and digital citizenship of the population is crucial in the information society.

Future research should focus on what is proper to the concept of digital citizenship: "critical awareness" and "digital activism", since the other dimensions are addressed by other conceptual categories and measurement tools such as "digital literacy" or "digital competence". The first dimension of digital citizenship "Critical Awareness" should combine the dimensions "Critical Perspective" and "Local/Global Awareness" of the CGC scale with some items of the other scales. Likewise, the second dimension of digital citizenship "Digital Activism" should combine the dimensions "Internet Political Activism" of the CGC scale with the "Online Civic Engagement" of the JM scale, incorporating the new forms of digital participation.

**Funding:** This research was funded by CentrA Foundation grant number PRY109/19 and supported by the R + D + I project (2020–2022) financed by the Government of Andalusia, grant number P18-RT-756 and by Ministry of Economic Affairs and Digital Transformation in PRODEBAT "PID2019-106254RB-I00".

**Institutional Review Board Statement:** Not applicable.

**Informed Consent Statement:** Not applicable.

**Data Availability Statement:** Fernández-Prados, J.S.; Lozano-Díaz, A. Comparative Study of Digital Citizenship Scales. Mendeley Data; 2021, V2, doi:10.17632/jdywv83dw4.2.

**Conflicts of Interest:** The authors declare no conflict of interest.

**Appendix A**

- Al-Zahrani, 2015 (AZ) [21]

*Dimension 1: Respect Yourself/Others*

1. I believe that everyone has basic digital rights, such as privacy and the right of expression and speech.

2. I believe that basic digital rights must be addressed, discussed, and understood by digital technology users.

3. I need to be taught about the inherent dangers of overuse of digital technologies.

4. I believe that creating destructive worms or viruses, creating Trojan Horses, and sending spam are digital crimes.

5. I understand the health and well-being risks surrounding the overuse of digital technologies, such as addiction and stress.

6. I believe that hacking into others' information, downloading illegal music and movies, plagiarizing, or stealing anyone's identification or property is unethical.

7. In an online digital environment, I always respect others' opinion and knowledge.

8. In an online digital environment, I always respect others' feelings.

9. In an online digital environment, I always make sure not to interrupt others when it is their turn.

10. I believe that digital technology users also have responsibilities, such as respecting others' basic digital rights.

11. I immediately delete emails from a suspicious source or sender.

12. When I feel unhappy or uncomfortable in an online digital environment, I try to express my feelings in a very rational way.

13. I use email service to communicate with others.

14. I believe in the importance of maintaining good physical and psychological health in this digital world.

15. I do not save any important information on public computers.

16. I believe that understanding digital rights and responsibilities helps everyone to be productive.

17. I believe that everyone should take responsibility for his/her online actions and deeds.

18. I believe that the use of digital technologies must be a compromise between earring and negligence.

19. Digital communication tools allow me to build new friendships in other parts of the world.

20. I have antivirus and Internet security protection on my computer.

21. I do not provide any unknown online parties with my personal information, such as bank accounts or credit cards.

22. In digital communication, I respect others' human rights, cultures, and right to expression.

23. Digital communication tools allow me to communicate with my friends easily.

24. In an online digital environment, I try to make sure that everyone has an equal opportunity for speech and discussion.

*Dimension 2: Educate Yourself/Others*

25. Electronic commerce gives me better choices.

26. Electronic commerce gives me more reasonable prices.

27. I always buy legal goods.

28. I do some research before buying anything from online stores.

29. Electronic commerce does not conflict with my society's regulations.

30. I love using electronic commerce tools (e.g., eBay and Amazon).

31. I prefer electronic commerce over going to the market.

32. I spend some time on social networks, such as Facebook and Twitter.

33. I use digital communication to express my opinion, learn, and share expertise.

34. I have been taught the new educational skills associated with digital technologies for the 21st century.

35. I only practice electronic commerce for goods that I cannot buy from or find in the market.

*Dimension 3: Protect Yourself/Others*

36. I always back up important data in a safe or external hard drive.

37. I always protect personal and important information in password-protected files.

38. I regularly change my passwords to protect my privacy.

39. I always read the privacy statement before installing new software.

40. I always do quick maintenance to remove unnecessary files and programs from my computer.

41. I have been taught about the possible threats when using new digital technologies.

42. I always visit trusted and harm-free websites.

43. When I notice strange things happening to my computer, I take it right away to the maintenance center.

44. I always find support when I encounter issues in using new digital technologies in my learning activities.

45. I have been trained on how to integrate new digital technologies in my future teaching activities.

46. I do not open any unknown or untrusted files.

- Jones and Mitchell, 2016 (JM) [23]

*Dimension 1: Online respect*

1. If I disagree with people online, I watch my language, so it doesn't come across as mean.

2. I am careful to make sure that the pictures I post or send of other people will not embarrass them or get them into trouble.

3. My favorite places to be online are where people are respectful toward each other.

4. I think about making sure that things I say and post online will not be something I regret later.

5. I do not add to arguments and insulting interactions that happen on the Internet.

6. I am careful about how I say things online, so they don't come across the wrong way.

7. I like to present myself online as someone making positive choices.

*Dimension 2: Online civic engagement*

1. I have used the Internet to improve my school or my town in some way.

2. I have used the Internet to learn how I can help a friend or help other kids in general.

3. When I am online, I try to end arguments or dramas when they develop.

4. I have used the Internet to share something that I am good at.

- Choi, Glassman, and Cristol, 2017 (CGC) [24]

*Dimension 1: Internet Political Activism*

1. I attend political meetings or public forums on local, town, or school affairs via online methods.

2. I work with others online to solve local, national, or global issues.

3. I organize petitions about social, cultural, political, or economic issues online.

4. I regularly post thoughts related to political or social issues online.

5. I sometimes contact government officials about an issue that is important to me via online methods.

6. I express my opinions online to challenge dominant perspectives or the status quo with regard to political or social issues.

7. I sign petitions about social, cultural, political, or economic issues online.

8. I work or volunteer for a political party or candidate via online methods.

9. I belong to online groups that are involved in political or social issues.

*Dimension 2: Technical Skills*

1. I can use the Internet to find information I need.

2. I can use the Internet to find and download applications (apps) that are useful to me.

3. I am able to use digital technologies (e.g., mobile/smart phones, Tablet PCs, Laptops, PCs) to achieve the goals I pursue.

4. I can access the Internet through digital technologies (e.g., mobile/smart phones, Tablet PCs, Laptops, PCs) whenever I want.

*Dimension 3: Local/Global Awareness (LGA)*

1. I am more informed with regard to political or social issues through using the Internet.

2. I am more aware of global issues through using the Internet.

*Dimension 4: Critical Perspective (CP)*

1. I think online participation is an effective way to make a change to something I believe to be unfair or unjust.

2. I think I am given to rethink my beliefs regarding a particular issue/topic when I use the Internet.

3. I think online participation is an effective way to engage with political or social issues.

4. I think online participation promotes offline engagement.

5. I think the Internet reflects the biases and dominance present in offline power structures.

6. I am more socially or politically engaged when I am online than offline.

7. I use the Internet in order to participate in social movement/change or protest.

*Dimension 5: Networking Agency (NA)*

1. Where possible, I comment on other people's writings in news websites, blogs, or SNSs I visit.

2. I enjoy communicating with others online.

3. I enjoy collaborating with others online more than I do offline.

4. I post original messages, audio, pictures, or videos to express my feelings/thoughts/ideas/opinions on the Internet.

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
