# Peer review of "Measuring Digital Citizenship: A Comparative Analysis"

_informatics, doi:10.3390/informatics8010018_

Round 1

Reviewer 1 Report

Although the different characteristics of Digital Citizenship is exactly the core intention of this paper, a broad first definition of the term should be given both inthe abstract and in the introduction.

The introduction already describes the state of the art in literature whereas the reader first should know what the paper is about.

The dimensions in Table 1 are very interesting but need an explanation and a discussion. Just mentioning the terms has no real value. It is a pity that the meaning of these dimensions is just listed in the annex.

In my opinion, just providing a statistical relevance of the various concepts of  a Digital Citizenship is of limited value.

What is the conclusion ? Should there be an internatinal standards for it ? How to get such a standard ? What would be its value ?

Minor English grammar errors:

- aims to show --> aims at showing

Author Response

REVIEWER 1

We are deeply grateful for the reviewer's contributions.

  • We have added a new table in the introduction with about ten definitions and critically commented on their evolution in the text. At the end of the introduction, we have tried to make clear the objective of our article.
  • We have included the table from the appendix in the text and developed the comments on it.
  • We have insisted somewhat more in the introduction and in the last section of the article on the theoretical debate behind the methodological issue.
  • We have underlined the usefulness of agreeing on a concept and methodology on digital citizenship to evaluate not only students but also the community and the population in the context of the information society.
  • We have corrected some errata and grammar errors.
    ...

Reviewer 2 Report

This submission aims to show a state of the art about digital citizenship from the methodological point of view when it comes to measuring this construct. The review of the scientific literature offers at least a dozen definitions and nine different scales of measurement. The comparative and dia-chronic analysis of the content of the definitions shows us two conceptions of digital citizenship, some more focused on digital competences and others on critical and activist aspects. This submission replicates and compares three scales of measurement of digital citizenship selected for their relevance and administered in a sample of 366 university students, in order to analyze both their psychometric properties and the existing coincidences and divergences between the three. 

The discussions look valuable. However,
1.This submission has not sufficiently clarified the novelty of the proposed survey. 
2. This submission misses discussing a few relevant works, such as
-"Digital Citizenship: You Can't Go Home Again",  TechTrends, vol 61, pp. 524–530, 2017

- “Information and Communications Technologies for Sustainable Development Goals: State-of-the-Art, Needs and Perspectives”, IEEE Communications Surveys & Tutorials, vol. 20, no. 3, pp. 2389-2406, March 2018 

-"Digital Rights, Digital Citizenship and Digital Literacy: What's the Difference?", Journal of New Approaches in Educational Research, 2021

- "Big Data Meet Green Challenges: Big Data toward Green Applications”, IEEE Systems Journal, vol. 10, no. 3, Sept. 2016 

Author Response

REVIEWER 2

We are deeply grateful for the reviewer's contributions.

We have found the text "Digital Rights, Digital Citizenship and Digital Literacy: What's the Difference?" especially helpful. We have incorporated this and other texts (i. g. “Information and Communications Technologies for Sustainable Development Goals: State-of-the-Art, Needs and Perspectives”) to clarify the sense of this work both in the introduction and in the last section.

Reviewer 3 Report

This paper replicates and compares three scales of measurement of digital citizenship selected for their relevance and administered in a sample of 366 university students, in order to analyze both their psychometric properties and the existing coincidences and divergences between the three

The main novelty of the paper should be clear and highlighted in the paper

The current challenges and issues should be mentioned

The experimental results should be explained in details, I would recommend to draw the results as graph to give better perception to reader

The methodology of the study is not clear

Author Response

REVIEWER 3

Very grateful for your comments and suggestions

  • We have further elaborated on the state of the art and highlighted at the end of the introduction the objective of this work.
  • We have included a final section on the lines and future research.
  • We have included subsections on methodology and results to present it more clearly.
  • We have included a figure and more explanations in the results.

Round 2

Reviewer 1 Report

Thank you for having taken my comments into account!

There is still some minor comment:

The paper would increase its value if it would not only urge for a harmonization of the different Digital Citizenship definitions, but if it would also pave the way and indicate what is common to all apporaches, where the differences lie and what would be the next steps.

Author Response

Dear Reviewer,
First of all thank you for your comment and according to your recommendation, we have added the following short paragraph in the last section of the article "Future Directions":

"Future research should focus on what is proper to the concept of digital citizenship: "critical awareness" and "digital activism", since the other dimensions are addressed by other conceptual categories and measurement tools such as "digital literacy" or "digital competence". The first dimension of digital citizenship "Critical Awareness" should combine the dimensions "Critical Perspective" and "Local/Global Awareness" of the CGC scale with some items of the other scales.  Likewise, the second dimension of digital citizenship "digital activism" should combine the dimensions "Internet Political Activism" of the CGC scale with the "Online civic engagement" of the JM scale, incorporating the new forms of digital participation."
